# ANALYZING THE EFFECTS OF EMULATING ON THE REINFORCEMENT LEARNING MANIFOLD

## ABSTRACT

Reinforcement learning has become a prominent research direction with the utilization of deep neural networks as state-action value function approximators enabling exploration and construction of functioning neural policies in MDPs with state representations in high dimensions. While reinforcement learning is currently being deployed in many different settings from medical to finance, the fact that reinforcement learning requires a reward signal from the MDP to learn a functioning policy can be restrictive for tasks in which the construction of the reward function is more or equally complex than learning it. In this line of research several studies proposed algorithms to learn a reward function or an optimal policy from observed optimal trajectories. In this paper, we focus on non-robustness of the state-of-the-art algorithms that accomplish learning without rewards in high dimensional state representation MDPs, and we demonstrate that the vanilla trained deep reinforcement learning policies are more resilient and value aligned than learning without rewards in MDPs with complex state representations.

## 1 INTRODUCTION

Learning from raw high dimensional state observations became possible with the utilization of effective function approximators (Mnih et al., 2015; 2016; Vinyals et al., 2019). Thus, this enhancement in the capabilities of reinforcement learning agents allowed these policies to be deployed in different fields from autonomous vehicles to language agents (OpenAI, 2023). One of the main limitations of reinforcement learning is to require a reward function to be able to learn an optimal policy for a given task. In some cases, constructing a reward function might be substantially more challenging than learning one, as in large language models (i.e. reinforcement learning from human feedback) (Glaese et al., 2022; Zhu et al., 2023). To solve this problem a line of research focused on learning the reward function solely based on observing expert trajectories, and several proposed to directly learn the optimal policy from the expert demonstrations. Even further, quite recent work demonstrated that learning from demonstrations is substantially more sample-efficient than deep reinforcement learning (Garg et al., 2021).

While deep reinforcement learning policies are vastly utilized in manifold settings, several concerns have been raised on the robustness and adversarial weaknesses of these deep neural policies (Huang et al., 2017; Gleave et al., 2020; Korkmaz, 2022; 2023). In particular, it has been shown that deep reinforcement learning policies can be manipulated via adversarial perturbations introduced to their state observations. Thus, the non-robustness of deep reinforcement learning limits the capabilities of policies trained in high dimensional state representation MDPs and raises safety concerns. Therefore, in our paper we seek answers to the following questions:

  i. *Does learning without rewards in high-dimensional state representation MDPs yield learning non-robust features independent from both the MDP and algorithm?*

  ii. *How do the state-of-the-art algorithms that focus on learning via emulating affect the policy robustness compared to straightforward vanilla training algorithms?*

  iii. *Does learning from expert demonstrations come with a cost compared to learning from exploration?*

Hence, to investigate these questions further in this paper we focus on the robustness and vulnerabilities of policies that can learn without a reward function towards adversarial formulations and make the following contributions:

- We investigate the robustness of the state-of-the-art deep neural policies that focus on learning via emulating against adversarial directions independent from both the algorithms the policies are trained with and the MDPs the policies are trained in. Our paper is the first one to focus on adversarial vulnerabilities of deep neural policies that can learn without a reward function.

- We theoretically motivate that learning from expert demonstrations comes with a great cost compared to learning from exploring. Learning without rewards causes sequential decision making processes to be extremely sensitive towards slight deviations from their optimal trajectories.

- We compare the robustness of straightforward vanilla trained policies and the policies that can learn without a reward function in high dimensional state representation MDPs. We demonstrate that vanilla trained deep reinforcement learning policies are significantly more robust compared to algorithms proposed to learn without the presence of a reward function (i.e. imitation learning and inverse reinforcement learning).

- Finally, we demonstrate that even extremely small divergence from optimal trajectories completely breaks the correlation between predicted rewards and true rewards obtained from the MDP for inverse deep neural policies.

## 2 RELEVANT WORK AND BACKGROUND

In reinforcement learning the environment is given by a Markov decision process (MDP) $\mathcal{M} = \{S, A, \mathcal{P}, p_0, r, \gamma\}$ where $S$ is the set of states, $A$ is the set of actions, $\mathcal{P}(s' \mid s, a)$ is the probability of transitioning to state $s'$ given that action $a$ is taken in state $s$, $p_0$ is the initial state distribution, $r(s, a)$ is the reward received when taking action $a$ in state $s$, and $0 < \gamma \leq 1$ is the discount factor. A policy $\pi(s, a)$ assigns a probability distribution on actions $a \in A$ to each state $s \in S$. Given a starting state distribution $p_0$, and transition probabilities $\mathcal{P}(\cdot \mid s_t, a_t)$, a policy $\pi$ defines a probability distribution $\mathbb{P}_\pi$ on trajectories $\{s_t, a_t\}_{t \geq 0}$ where $s_0 \sim p_0$, $a_t \sim \pi(s_t, \cdot)$, and $s_{t+1} \sim \mathcal{P}(\cdot \mid s_t, a_t)$. In particular, the distribution $\mathbb{P}_\pi$ satisfies $\mathbb{P}_\pi[s_0 = s] = p_0(s)$,

$$\mathbb{P}_\pi[a_t = a \mid s_t = s] = \pi(s, a), \mathbb{P}_\pi[s_{t+1} = s' \mid s_t = s, a_t = a] = \mathcal{P}(s' \mid s, a).$$

The goal in reinforcement learning is to learn a policy $\pi(s, a)$ that maximizes the expected discounted cumulative rewards $\sum_t \gamma^t \mathbb{E}_{s_t, a_t \sim \mathbb{P}_\pi}[r(s_t, a_t)]$. The occupancy measure $\rho_\pi$ for a policy $\pi$ is the distribution over states and actions visited when executing the policy given by $\rho_\pi(s, a) = \pi(s, a) \sum_t \gamma^t \mathbb{P}_\pi[s_t = s]$. In soft-$Q$ learning the policy is determined by learning the soft-$Q$ function $Q(s, a)$. Given a policy $\pi$ and a function $Q(s, a)$ the soft value function is given by $\mathcal{V}^\pi(s) = \mathbb{E}_{a \sim \pi(s, \cdot)}[Q(s, a) - \log \pi(s, a)]$, and the soft Bellman operator is

$$(\mathcal{T}^\pi Q)(s, a) = r(s, a) + \gamma \mathbb{E}_{s' \sim \mathcal{P}(\cdot \mid s, a)}[\mathcal{V}^\pi(s', a)].$$

The soft Bellman operator is contractive and defines a unique soft-$Q$ function satisfying the soft Bellman equation $Q = \mathcal{T}^\pi Q$. In soft-$Q$ learning the goal is to learn a policy $\pi$ which maximizes the entropy-regularized reward $\sum_t \gamma^t \mathbb{E}_{s_t, a_t \sim \mathbb{P}_\pi}[r(s_t, a_t) - \log(\pi(s_t, a_t))]$. The optimal policy is given by $\pi(s, a) = \frac{\exp Q(s, a)}{\sum_{a'} \exp Q(s, a')}$, where $Q$ is the soft-$Q$ function satisfying the soft Bellman equation $Q(s, a) = r(s, a) + \gamma \mathbb{E}_{s' \sim \mathcal{P}(\cdot \mid s, a)}[\log \sum_{a'} Q(s', a')]$.

**Robustness in Deep Reinforcement Learning:** The first work focusing on adversarial robustness of deep reinforcement learning policies was conducted by Huang et al. (2017). In particular, the study conducted by Huang et al. (2017) utilizes the fast gradient sign method produced perturbations (introduced by Goodfellow et al. (2015)) added to the state observations of the deep reinforcement learning policies. On this line of research while some studies focus on investigating more efficient ways of producing these adversarial perturbations introduced to state observations, some utilize adversarial formulations (Carlini & Wagner, 2017) to investigate frequency vulnerabilities and visualize the different patterns of non-robust features learnt by different deep reinforcement learning

algorithms. Along these lines, some studies investigate the relationship between adversarial directions and natural directions intrinsic to the MDP and demonstrate that adversarial training results in worse generalization capabilities compared to vanilla training (Korkmaz, 2023). Following the research focusing on investigating vulnerabilities of deep reinforcement learning policies towards adversarial perturbations several other works focus on different strategies to make policies more robust towards these malicious perturbations. More in particular, Gleave et al. (2020); Pinto et al. (2017) propose to formulate the relationship between the adversary and the policy as a zero-sum Markov game. In some of these studies the adversary is restricted to change environment dynamics (Pinto et al., 2017), and in others the adversary is restricted to taking natural actions in the given environment (Gleave et al., 2020). Some studies focused on detection of these adversarial (i.e. non-robust) state observations by leveraging the curvature of the deep reinforcement learning manifold to make robust decisions (Korkmaz & Brown-Cohen, 2023). While several studies focused on trying to build robust deep reinforcement learning policies, quite recently several studies demonstrated that deep reinforcement learning policies learn shared adversarial features across MDPs including the state-of-the-art adversarially trained ones (Korkmaz, 2022; 2021).

**Learning without Rewards:** Reinforcement learning from human feedback (RLHF) resulted in substantial progress in large language models (OpenAI, 2023). To ensure safety and aligned values with human preferences, currently reinforcement learning from human feedback is the main method used widely in large language models (Glaese et al., 2022; Zhu et al., 2023; Menick et al., 2022). Quite recent work demonstrated the connection between learning from preferences and inverse reinforcement learning, and further provided sample complexity bounds for these algorithms (Zhu et al., 2023). Note that inverse reinforcement learning focuses on learning a reward function from a set of expert trajectory observations. Hence, upon the construction of a reward function from observations an optimal policy can be learnt via straightforward reinforcement learning. Another line of research that centers on learning without a reward function focuses on the setting of learning a functioning policy from a given set of observed expert trajectories via emulating expert behaviour. Quite recently, Garg et al. (2021) proposed to learn a single Q function from expert demonstrations to both represent the reward function and the policy. The proposal of learning a soft-Q function is currently the only algorithm that can achieve a performance level that can match deep reinforcement learning in high dimensional state representation MDPs. Furthermore, the authors argue that the fact that inferred rewards are highly correlated with the ground truth rewards shows that the proposed algorithms can also be used in inverse reinforcement learning. For this reason in the remainder part of the paper we will refer to inverse reinforcement and imitation learning policies as inverse deep neural policies.

## 3 THE OUTCOMES OF LACK OF EXPLORATION IN LEARNING WITHOUT REWARDS

In this section we fundamentally explain and theoretically motivate the results observed and reported in Section 5. Thus, this section is dedicated to explain the natural cases where training via the state-of-the-art deep imitation learning algorithms leads to policies that are non-robust. Let each state be given by a $d$-dimensional feature vector $s \in \mathbb{R}^d$, and for each action $a \in A$ there is a parameter vector $\theta_a \in \mathbb{R}^d$. The state-action value function is parameterized as $Q_\theta(s, a) = \langle \theta_a, s \rangle$. The inverse $Q$-learning objective is given by

$$\mathbb{E}_{(s,a)\sim\rho_E} \left[ \phi \left( Q_\theta(s, a) - \gamma \mathbb{E}_{s'\sim\mathbb{P}(\cdot|s,a)}[\mathcal{V}_\theta(s')] \right) \right] \tag{1}$$
$$- \mathbb{E}_{(s,a)\sim\mu} \left[ \phi \left( \mathcal{V}_\theta(s) - \gamma \mathbb{E}_{s'\sim\mathbb{P}(\cdot|s,a)}[\mathcal{V}_\theta(s')] \right) \right]$$

where $\mathcal{V}_\theta(s) = \log \sum_a \exp Q_\theta(s, a)$, and let $\rho_E$ be the occupancy measure of the expert policy, and $\mu$ any valid occupancy measure. We will focus on the offline setting where $\mu = \rho_E$ i.e. only samples from expert trajectories are used. Later we will discuss extensions to the online setting where $\mu$ is a mixture of expert trajectories and previously sampled states from a replay buffer. Expert trajectories will achieve higher rewards than the trajectories generated by an agent early on in the training process. Thus, we show that this corresponds to states in expert trajectories having larger projection onto a low-dimensional subspace.

**Lemma 3.1.** *Let $\theta^*$ be the parameters of the optimal soft-Q function $Q^*(s, a) = \langle \theta_a^*, s \rangle$, and let $\mathcal{V}_{\theta^*}$ be the corresponding soft value function. Let $B > 1$ and $s, s' \in \mathbb{R}^d$ be states such that*

$\mathcal{V}_{\theta^*}(s) > B\mathcal{V}_{\theta^*}(s')$. *Then there is a subspace $W \subseteq \mathbb{R}^d$ of dimension at most $|A|$ and a linear transformation $L : \mathbb{R}^d \to \mathbb{R}^{|A|}$ with kernel $W^\perp$ such that*

$$\|Ls\| > \frac{B}{\sqrt{|A|}}\|Ls'\| - \log|A|$$

*Proof.* Let $W$ be the span of $\theta_a^*$ for $a \in A$. Let $L$ to be the linear map defined by the matrix with rows given by $\theta_a^*$. Clearly the kernel of $L$ is the orthogonal complement $W^\perp$. Recalling that $\mathcal{V}_{\theta^*}(s) = \log \sum_a \exp Q_{\theta^*}(s, a)$ we have

$$\mathcal{V}_{\theta^*}(s') \geq \max_{a \in A} Q_{\theta^*}(s', a) \geq \frac{1}{\sqrt{|A|}}\sqrt{\sum_{a \in A} Q_{\theta^*}(s', a)^2} = \frac{1}{\sqrt{|A|}}\|Ls'\|.$$

On the other hand

$$\mathcal{V}_{\theta^*}(s) \leq \max_{a \in A} Q_{\theta^*}(s, a) + \log|A| \leq \sqrt{\sum_{a \in A} Q_{\theta^*}(s, a)^2} + \log|A| = \|Ls\| + \log|A|.$$

Combining the two inequalities with the assumption that $\mathcal{V}_{\theta^*}(s) > B\mathcal{V}_{\theta^*}(s')$ completes the proof. $\square$

Lemma 3.1 states that for $B \gg \sqrt{|A|}$, the projection onto $W$ of $s$ is much larger than that of $s'$ (as measured by the transformation $L$). Thus, the intuitive conclusion is that states with higher value (e.g. states sampled from expert trajectories) have larger projection onto a low-dimensional subspace. The inverse $Q$-learning algorithm updates the parameters $\theta$ by stochastic gradient descent, where the average over $\rho_E$ in Equation 1 is approximated by sampling from stored expert trajectories. In order to qualitatively demonstrate how the use of expert trajectories leads to lower robustness we take the conclusion of Lemma 3.1 and define a model for this setting which we refer to as subspace-contained expert trajectories.

**Definition 3.2.** The set of expert trajectories $\rho_E$ is *subspace-contained* if there exists a subspace $W \subseteq \mathbb{R}^d$ of dimension $l < d$ such that, when started in $s_0 \in W$ the expert policy $\pi_E$ always transitions to states $s \in W$. Consequently, $s \in W$ for all $(s, a)$ in the support of $\rho_E$.

While Lemma 3.1 shows that in general there will exist a subspace where higher value states have larger projection than lower value states, Definition 3.2 posits the existence of a potentially lower-dimensional subspace which entirely contains all the states encountered on expert trajectories. Due to the fact that expert trajectories will tend to have higher values, Definition 3.2 is a natural strengthening of Lemma 3.1 that furthermore enables a clean theoretical analysis of the effects of inverse $Q$-learning. In particular, we show that the $Q$-values for states in the orthogonal complement of $W$ will remain unchanged during training with inverse $Q$-learning.

**Proposition 3.3.** *Let the initial weights be $\theta^{(0)} \in \mathbb{R}^d$. Let $\rho_E$ be subspace-contained, and let $s \in W^\perp$ be a state in the orthogonal complement of $W$. Let $\theta$ be the weights after any number of steps of training with inverse $Q$-learning. Then $Q_\theta(s, a) = Q_{\theta^{(0)}}(s, a)$ for all $a \in A$.*

*Proof.* We first show that the (stochastic) gradient of the cost function in Equation 1 is contained in $W$. For any $s$ the gradient of the state-action value function is

$$\nabla_{\theta_a} Q_\theta(s, a) = \nabla_{\theta_a}\langle\theta_a, s\rangle = s, \quad \nabla_{\theta_a} Q_\theta(s, a') = \nabla_{\theta_a}\langle\theta_{a'}, s\rangle = 0.$$

Further, the gradient of the value function is

$$\nabla_{\theta_a}\mathcal{V}_\theta^*(s) = \nabla_{\theta_a}\log\sum_a \exp Q_\theta(s, a) = s\frac{\exp Q_\theta(s, a)}{\sum_{a'}\exp Q_\theta(s, a')}.$$

Note that all the above gradients are multiples of the input state $s$. Next observe that the $a$ component gradient of the objective in Equation 1 is a linear combination of $\nabla_{\theta_a} Q_\theta(s, a)$, $\nabla_{\theta_a} Q_\theta(s, a')$, and $\nabla_{\theta_a}\mathcal{V}_\theta^*(s)$, where $s$ always lies in the support of $\rho_E$. Thus, by Definition 3.2, the gradient update in each step of inverse $Q$-learning lies in the subspace $W$, as it is a linear combination of states $s \in W$.

Therefore, after any number of steps the weights $\theta$ will satisfy $\theta_a - \theta_a^{(0)} \in W$. To complete the proof observe that for $s \in W^\perp$ we have $Q_\theta(s, a) - Q_{\theta^{(0)}}(s, a) = \langle\theta_a - \theta_a^{(0)}, s\rangle = 0$. $\square$

The initial weights $\theta^{(0)}$ are usually set randomly or to some default value, and thus one does not expect $\theta^{(0)}$ to have any reasonable relationship to the optimal weights. Therefore, the intuitive consequence of Proposition 3.3 is that the soft $Q$-function learned from expert trajectories assigns random or arbitrary values to actions in states $s \in W^\perp$. In fact, Proposition 3.3 leads directly to the following corollary which shows that the rewards estimated by inverse $Q$-learning become uncorrelated to the true rewards.

**Corollary 3.4.** *Suppose each coordinate of $\theta^{(0)}$ is an independent Gaussian random variable with mean $0$ and variance $1$, and that the feature vectors corresponding to each state are normalized i.e. $\|s\|_2 = 1$. Then for each $s \in W^\perp$ the expectation over $\theta^{(0)}$ of the rewards $r_\theta(s, a)$ estimated by inverse $Q$-learning will be independent of the state $s$.*

*Proof.* The reward estimates for inverse reinforcement learning are obtained by assuming that the learned $Q$-function satisfies the soft Bellman equation i.e. that $Q_\theta(s, a) = r(s, a) + \mathbb{E}_{s' \sim \mathcal{P}(\cdot|s,a)}[\mathcal{V}^{\pi_\theta}(s')]$. Thus, given the estimated optimal state-action value function $Q_\theta$, one solves for the rewards $r(s, a)$ in the soft Bellman equation in order to obtain the reward estimate. Hence, the reward estimated by inverse $Q$-learning in a state $s \in W^\perp$ is

$$r_\theta(s, a) = Q_\theta(s, a) - \mathbb{E}_{s' \sim \mathcal{P}(\cdot|s,a)}[\mathcal{V}^{\pi_\theta}(s')] = Q_\theta(s, a) - \mathbb{E}_{s' \sim \mathcal{P}(\cdot|s,a)}[\log \sum_{a'} \exp Q_\theta(s', a')]$$

$$= Q_{\theta^{(0)}}(s, a) - \mathbb{E}_{s' \sim \mathcal{P}(\cdot|s,a)}[\log \sum_{a'} \exp Q_{\theta^{(0)}}(s', a')]$$

where the last line follows from Proposition 3.3. Because $\theta^{(0)}$ has independent Gaussian coordinates, its distribution is rotationally invariant i.e. $U\theta_a^{(0)}$ has the same distribution as $\theta_a^{(0)}$ for any rotation matrix $U$. Let $U$ be any rotation such that $U^\top$ sends $s$ to the first standard basis vector $e_1$. It is always possible to choose such a rotation because $\|s\| = 1$. Then by rotational invariance $Q_{\theta^{(0)}}(s, a) = \langle \theta_a^{(0)}, s \rangle$ has the same distribution as $\langle U\theta_a^{(0)}, s \rangle = \langle \theta_a^{(0)}, U^\top s \rangle = \langle \theta_a^{(0)}, e_1 \rangle = Q_{\theta^{(0)}}(e_1, a)$. Thus the expectation of the rewards estimated by inverse $Q$-learning is given by

$$\mathbb{E}_{\theta^{(0)} \sim \mathcal{N}(0,I)}[r_\theta(s, a)] = \mathbb{E}_{\theta^{(0)} \sim \mathcal{N}(0,I)}[Q_{\theta^{(0)}}(s, a)]$$

$$- \mathbb{E}_{s' \sim \mathcal{P}(\cdot|s,a)}\left[\mathbb{E}_{\theta^{(0)} \sim \mathcal{N}(0,I)}\left[\log \sum_{a' \in \mathcal{A}} \exp Q_{\theta^{(0)}}(s', a')\right]\right]$$

$$= \mathbb{E}_{\theta^{(0)} \sim \mathcal{N}(0,I)}[Q_{\theta^{(0)}}(e_1, a)] - \mathbb{E}_{s' \sim \mathcal{P}(\cdot|s,a)}\left[\mathbb{E}_{\theta^{(0)} \sim \mathcal{N}(0,I)}\left[\log \sum_{a' \in \mathcal{A}} \exp Q_{\theta^{(0)}}(e_1, a')\right]\right]$$

$$= \mathbb{E}_{\theta^{(0)} \sim \mathcal{N}(0,I)}[Q_{\theta^{(0)}}(e_1, a)] - \mathbb{E}_{\theta^{(0)} \sim \mathcal{N}(0,I)}\left[\log \sum_{a' \in \mathcal{A}} \exp Q_{\theta^{(0)}}(e_1, a')\right].$$

This completes the proof as the above expression for $\mathbb{E}_{\theta^{(0)} \sim \mathcal{N}(0,I)}[r_\theta(s, a)]$ does not depend on the state $s$. $\square$

In general, the inverse $Q$-learning policy still may perform well. Indeed, if following the learned policy causes the agent to only encounter states $s \in W$, then performance in the standard setting will be unaffected by inaccuracy in states $s \in W^\perp$. However, the robustness of the policy may still be affected, as a slight deviation from the optimal path may cause the policy to transition into $s \in W^\perp$ where the $Q$-function is completely untrained. We now formalize the above intuition regarding the consequences of Proposition 3.3. Let $\pi^*$ be the optimal soft policy and let $Q_\theta$ be the soft $Q$-function obtained by training with inverse $Q$-learning with corresponding soft policy $\pi_\theta$. We assume that $\pi_\theta(s, a) = \pi^*(s, a)$ for all $s \in W$ i.e. training with inverse $Q$-learning has succeeded in accurately learning the optimal soft policy in $W$. The optimal policy $\pi^*$ started at $s_0 \in S$ never transitions out of $S$. However, $\pi^*$ receives the same expected cumulative rewards $R$ when started in either $s_0 \in W$ or $s' \in W^\perp$. Taking non-optimal actions in a $p$ fraction of states $s \in W$ and utilizing $\pi^*$ in all other states results in a transition to $s' \in W^\perp$. There are no transitions from $s' \in W^\perp$ to $s \in W$. We now show that under these assumptions, slight deviations from the optimal policy have no impact on the optimal policy $\pi^*$, but cause the inverse deep neural policy $\pi_\theta$ to perform at the same level as an untrained policy.

Table 1: The performance drop results with algorithm and MDP independent adversarial direction $\mathcal{A}_{\text{alg}+\mathcal{M}}^{\text{random}}$ and MDP independent adversarial direction $\mathcal{A}_{\mathcal{M}}^{\text{random}}$ for vanilla trained deep reinforcement learning policy and inverse deep neural policy in Pong.

| Training Method | Adversarial Setting | RoadRunner | BankHeist | TimePilot |
|---|---|---|---|---|
| Deep Inverse-$Q$ Learning | $\mathcal{A}_{\text{alg}+\mathcal{M}}^{\text{random}}$ | 1.0±0.0 | 0.9153±0.01128 | 0.9880±0.0112 |
| Vanilla Trained | $\mathcal{A}_{\mathcal{M}}^{\text{random}}$ | 0.0819±0.0500 | 0.0265±0.01284 | 0.2963±0.0573 |

| Training Method | Adversarial Setting | JamesBond | CrazyClimber | Gaussian |
|---|---|---|---|---|
| Deep Inverse-$Q$ Learning | $\mathcal{A}_{\text{alg}+\mathcal{M}}^{\text{random}}$ | 1.0±0.0 | 0.9857±0.006 | 0.04285±0.01572 |
| Vanilla Trained | $\mathcal{A}_{\mathcal{M}}^{\text{random}}$ | -0.0024±0.0037 | 0.06024±0.02970 | 0.0451±0.0182 |

**Proposition 3.5.** *Taking non-optimal actions in a $p$ fraction of states does not change the reward received by the optimal policy $\pi^*$, but causes the inverse reinforcement learning policy $\pi_\theta$ to receive the same rewards received by $\pi_{\theta^{(0)}}$.*

*Proof.* After taking non-optimal actions in a $p$ fraction of states, the optimal policy $\pi^*$ transitions to $s' \in W^\perp$, and by assumption never transitions out. However, $\pi^*$ receives the same rewards after transitioning to $s' \in W^\perp$ as it would have from remaining in $W$. The policy $\pi_\theta$ is equal to $\pi^*$ on $W$. Thus, taking non-optimal actions in a $p$ fraction of states causes a transition to $W^\perp$. However, by Proposition 3.3 $\pi_\theta(s', a) = \pi_{\theta^{(0)}}(s', a)$ for all $s \in W^\perp$. By assumption there are no transitions out of $W^\perp$, so $\pi_\theta$ receives the same rewards as $\pi_{\theta^{(0)}}$. $\square$

# 4 PROBING THE INVERSE REINFORCEMENT LEARNING MANIFOLD VIA ADVERSARIAL DIRECTIONS

Section 3 is dedicated to provide theoretically motivated fundamental reasoning behind the non-robustness of the deep inverse $Q$-learning algorithm laid out in Section 4 and in Section 5. Hence, in this section we describe the methods used to cause deviations from the optimal trajectory in order to understand the robustness of policies trained with inverse reinforcement learning. To achieve this there are two main approaches we will take, one is based on moving along the adversarial directions in the deep neural policy manifold, and the second is to directly slightly push the policy from its optimal course of trajectory. These are described below in more detail. For the adversarial directions we utilize the methodology described in Korkmaz (2022). Precisely,

**Definition 4.1.** *Algorithm and MDP independent adversarial direction $\mathcal{A}_{\text{alg}+\mathcal{M}}^{\text{random}}$*: Given a random state $s(\mathcal{M})$ sampled from a random episode of $e$ of an MDP $\mathcal{M}$ from a policy $\pi(s(\mathcal{M}), \cdot)$, the minimum length adversarial direction $\boldsymbol{v}(s(\mathcal{M}), \pi(s(\mathcal{M}), \cdot))$ is computed satisfying, $\arg\max_a \pi(s(\mathcal{M}), a) \neq \arg\max_a \pi(s(\mathcal{M}) + \boldsymbol{v}(s(\mathcal{M}), \pi(s(\mathcal{M}), \cdot)), a)$. The computed adversarial direction $\boldsymbol{v}(s(\mathcal{M}), \pi(s(\mathcal{M}), \cdot))$ norm-bounded by $\kappa > 0$ is added to the visited states of another policy $\pi'(s(\mathcal{M}'), \cdot))$ trained with a completely different algorithm, in a distinct MDP $\mathcal{M}'$. Hence, the state obtained moving along the adversarial direction is

$$s_{\boldsymbol{v}} = s(\mathcal{M}') + \kappa \frac{\boldsymbol{v}(s(\mathcal{M}), \pi(s(\mathcal{M}), \cdot))}{\|\boldsymbol{v}(s(\mathcal{M}), \pi(s(\mathcal{M}), \cdot))\|}.$$

Note that if $\mathcal{M}_{\text{alg}} \neq \mathcal{M}'_{\text{alg}}$ this means that the policies are trained with the same algorithm but in distinct MDPs; thus, the adversarial direction is computed from $\mathcal{M}$ and transferred to a distinct MDP $\mathcal{M}'$. The setting of $\mathcal{M}_{\text{alg}} \neq \mathcal{M}'_{\text{alg}}$ will be referred as $\mathcal{A}_{\mathcal{M}}^{\text{random}}$.

**Definition 4.2.** *The $\delta$-deviation from the optimal trajectory*: For a policy $\pi(s, a)$ and state $s$ let $a_w(s) = \arg\min_a Q(s, a)$ denote the worst action in state $s$. The notation $\mathcal{A}_{a_w}$ refers to a setting in which in state $s$ the policy is set to take action $a_w(s)$, rather than the optimal action selected by the policy $\pi(s, a)$ for a $\delta$-fraction of the visited states where delta is $\delta \ll 1$. The notation $\mathcal{A}_{\text{random}}$ refers to the setting in which the policy $\pi(s, a)$ is set to take an action uniformly at random $a \sim \mathcal{U}_A$ in $s$ for a $\delta$-fraction of the visited states where delta is $\delta \ll 1$.

Both Definition 4.1 and Definition 4.2 will be used in Section 5 to lay out precise non-robustness of inverse deep neural policies and their comparison to vanilla trained deep reinforcement learning

Table 2: The performance drop results with algorithm and MDP independent adversarial direction $\mathcal{A}_{\text{alg}+\mathcal{M}}^{\text{random}}$ and MDP independent adversarial direction $\mathcal{A}_{\mathcal{M}}^{\text{random}}$ for vanilla trained deep reinforcement learning policies and inverse deep neural policies in Seaquest.

| Training Method | Adversarial Setting | RoadRunner | BankHeist | TimePilot |
|---|---|---|---|---|
| Deep Inverse-$Q$ Learning | $\mathcal{A}_{\text{alg}+\mathcal{M}}^{\text{random}}$ | 0.93439±0.0075 | 0.9882±0.03457 | 0.95037±0.01454 |
| Vanilla Trained | $\mathcal{A}_{\mathcal{M}}^{\text{random}}$ | 0.28496±0.09693 | 0.18527±0.14817 | 0.46938±0.08672 |
| Training Method | Adversarial Setting | JamesBond | CrazyClimber | Gaussian |
| Deep Inverse-$Q$ Learning | $\mathcal{A}_{\text{alg}+\mathcal{M}}^{\text{random}}$ | 0.916947±0.01371 | 0.76282±0.02146 | 0.05277±0.0821 |
| Vanilla Trained | $\mathcal{A}_{\mathcal{M}}^{\text{random}}$ | 0.308188±0.12931 | 0.180897±0.13982 | 0.05536±0.12091 |

policies. Note that the initial and fundamental objective of inverse reinforcement learning on inferring a reward function from observed trajectories opened a new channel on the value alignment problem Ng & Russell (2000); Russell (1998). More precisely, the current unethical behaviour observed in large language models that are trained with RLHF (i.e. learning a reward function from human preferences to align language agents with human values) is one of the concrete substantial safety concerns that is tightly connected to what we highlight in our paper. Thus, the vulnerabilities described in Section 5 carry critical importance due to the fact that the results reported demonstrate that simpler algorithms such as vanilla deep reinforcement learning are more robust compared to algorithms that are specifically focused on resolving the value-alignment problem. In particular, the fact that the policy's perception on the environmental (i.e. ground truth) rewards is broken as reported in Section 5.2 further demonstrates that the value-alignment problem is far beyond being solved, and in fact most critically, the claimed alignment is extremely fragile.

## 5 THE COST OF LEARNING FROM EXPERT DEMONSTRATIONS

In this paper the straightforward vanilla trained deep reinforcement learning policies are trained with Deep Double Q-Network (DDQN) initially proposed by Hasselt et al. (2016) with the architecture introduced in Wang et al. (2016). The state-of-the-art imitation and inverse reinforcement learning policy is trained via the inverse $Q$-learning algorithm described in Section 2. The experiments are conducted in the Arcade Learning Environment (ALE) Bellemare et al. (2013) with OpenAI wrappers Brockman et al. (2016). The results are averaged over 10 episodes and the standard error of the mean is included in all the tables and figures presented in the paper. The normalized performance drop of the policies is computed as $\mathcal{P} = (\text{Score}_{\text{max}} - \text{Score}_{\text{set}})/(\text{Score}_{\text{max}} - \text{Score}_{\text{min}})$. Here $\text{Score}_{\text{max}}$ is the score obtained by the baseline trained policy following the learned policy with a clean run in a given environment, $\text{Score}_{\text{set}}$ is the score obtained by the policy in the test time, and $\text{Score}_{\text{min}}$ is the score obtained by the trained policy when the policy chooses the worst possible action in each state. Scores represent the cumulative rewards obtained by the policy, and are recorded at the end of an episode. Note that $\mathcal{A}_{\text{base}}$ refers to the unmodified run of the policy in an unmodified MDP.

As described in detail in Section 2 the inverse $Q$-learning algorithm learns both an optimal policy and a reward function from observed trajectories; thus, the fact that inverse-$Q$ learning simultaneously learns both a reward function and an optimal policy is the reason that throughout the paper imitation learning and inverse reinforcement learning will be used interchangeably. Note that the inverse $Q$-learning algorithm is the only algorithm that can achieve equivalent performance with vanilla trained deep reinforcement learning policies in MDPs with high-dimensional observations. Table 1 and Table 2 report the performance drop results with the environment and algorithm independent random state adversary $\mathcal{A}_{\text{alg}+\mathcal{M}}^{\text{random}}$ and the environment independent random state adversary $\mathcal{A}_{\mathcal{M}}^{\text{random}}$ for vanilla trained deep reinforcement learning policies and inverse deep neural policies in Arcade Learning Environment (ALE). Recall that $\mathcal{A}_{\text{alg}+\mathcal{M}}^{\text{random}}$ represents the adversarial setting in which the adversarial direction is computed from a completely different MDP and from a completely different trained policy. Note that in these experiments the $\ell_2$-norm bound $\kappa$ level is set to the magnitude where simple Gaussian noise with the $\ell_2$-norm $\kappa$ has insignificant effect on the policy performance.

In particular, the results reported in Table 1 and 2 are for an adversarial direction that is computed from a vanilla trained policy in one MDP $\mathcal{M}$, and added to the observations in a different MDP $\mathcal{M}'$ of the inverse $Q$-learning policy and the vanilla trained policy respectively. This corresponds to the $\mathcal{A}_{\text{alg}+\mathcal{M}}^{\text{random}}$ setting for the inverse $Q$-learning policy, and $\mathcal{A}_{\mathcal{M}}^{\text{random}}$ for the vanilla policy. Of particular

Table 3: Pearson and Spearman correlation coefficient between true cumulative rewards obtained from the environment and cumulative reward prediction made by the state-of-the-art inverse deep neural policy for a baseline run $\mathcal{A}_{\text{base}}$, with $\mathcal{A}_{a_w}$, $\delta$-deviation $\mathcal{A}_{\text{random}}$, and with $\mathcal{A}_{\text{alg}+\mathcal{M}}^{\text{random}}$ where $\delta = 0.003$.

| Seaquest | $\mathcal{A}_{\text{base}}$ | $\delta$-deviation $\mathcal{A}_{a_w}$ | $\delta$-deviation $\mathcal{A}_{\text{random}}$ | $\mathcal{A}_{\text{alg}+\mathcal{M}}^{\text{random}}$ |
|---|---|---|---|---|
| Pearson | 0.857880±0.025528 | -0.44787±0.140356 | 0.324616±0.22191 | 0.202436±0.24088 |
| Spearman | 0.688300±0.1257206 | -0.35129±0.20131 | 0.023051±0.12057 | 0.023701±0.20764 |
| BeamRider | $\mathcal{A}_{\text{base}}$ | $\delta$-deviation $\mathcal{A}_{a_w}$ | $\delta$-deviation $\mathcal{A}_{\text{random}}$ | $\mathcal{A}_{\text{alg}+\mathcal{M}}^{\text{random}}$ |
| Pearson | 0.654375±0.005274 | -0.27414±0.026496 | -0.10209±0.076554 | -0.22879±0.199385 |
| Spearman | 0.688300±0.1257206 | -0.34725±0.022020 | -0.11846±0.092472 | -0.43719±0.115118 |
| Breakout | $\mathcal{A}_{\text{base}}$ | $\delta$-deviation $\mathcal{A}_{a_w}$ | $\delta$-deviation $\mathcal{A}_{\text{random}}$ | $\mathcal{A}_{\text{alg}+\mathcal{M}}^{\text{random}}$ |
| Pearson | 0.850592±0.013398 | -0.12104±0.083332 | 0.248250±0.046503 | -0.09089±0.092005 |
| Spearman | 0.7665362±0.04679 | -0.13360±0.030187 | 0.243274±0.012255 | -0.25481±0.089637 |

significance is the fact that, even though the perturbation is specifically computed from a vanilla trained policy and for a vanilla trained policy, the impact of the adversarial direction is much larger on the inverse-$Q$ learning policy than on the vanilla policy. Thus, the results in Table 1 and 2 demonstrate that the inverse $Q$-learning policies are more susceptible to the shared adversarial directions across both MDPs and algorithms compared to vanilla trained reinforcement learning policies.

### 5.1 THE CATASTROPHIC RESULTS OF SMALL DIVERGENCE FROM THE OPTIMAL POLICY

In this section we investigate effects of small deviations from the optimal policy followed by the inverse deep neural policy. To achieve this, we will use $\mathcal{A}_{a_w}$ with $a_w = \arg\min_{a' \in \mathcal{A}} Q(s, a')$ for a random small fraction of visited states in a given episode, the $\mathcal{A}_{\text{random}}$ setting explained in Definition 4.2, and the $\mathcal{A}_{\text{alg}+\mathcal{M}}^{\text{random}}$ setting explained in Definition 4.1. Either moving along adversarial directions towards the non-robust region in the deep neural policy manifold or concrete changes in the actions taken by the policy will cause slight deviation from the optimal course of the inverse deep neural policy. Figure 1 reports the true rewards obtained from the environment and the reward predictions of inverse $Q$-learning, the Pearson correlation coefficient between inverse $Q$-learning reward predictions and environment rewards, and the performance drop computed from environment rewards and the inverse $Q$-learning predictions with respect to $\delta$-deviation from optimal trajectory with $\mathcal{A}_{a_w}$ and $\mathcal{A}_{\text{random}}$. Thus, the results in Figure 1 demonstrate the outcome of slight deviation of the optimal trajectory and its effects on the reward predictions of the inverse $Q$-learning policy and the true rewards obtained from the environment. The fact that small deviations from the optimal policy result in significant decrease in both the rewards obtained, and the accuracy of the predicted rewards for the inverse deep neural policy, provide empirical verification of Corollary 3.4 given as a theoretical justification in Section 3. In particular, this lends credence to the claim that the lower exploration of state space and being limited to expert trajectories cause the inverse deep neural policies to overfit to the expert's beliefs and experience significant performance loss under subtle departures from the policy's optimal course. Notably, when $\delta$ is initially increased from zero, the performance drop of the inverse $Q$-learning reward predictions is negative. That is, despite a decrease in performance caused by the deviation, the inverse $Q$-learning policy believes it will actually receive larger rewards than it did before. In fact, as the true environment rewards decrease, their Pearson correlation with the predicted rewards also decreases, indicating that in general small deviations cause the policy to form an inaccurate view of the rewards it will obtain. Overall, while the cost of learning from expert demonstrations instead of exploration is demonstrated in Table 1, the fact that during training the inverse Q-learning policy is not exposed to a more diverse set of observations in the state space is the foundational reason for the results observed in Figure 1.

### 5.2 BREAKING THE LINK BETWEEN IMITATION AND INVERSE REINFORCEMENT LEARNING

Table 3 shows the Pearson and Spearman correlation coefficients between cumulative rewards obtained from the environment and the cumulative reward prediction of the deep imitation learning policy for the algorithm and MDP independent adversarial direction $\mathcal{A}_{\text{alg}+\mathcal{M}}^{\text{random}}$, $\delta$-deviation $\mathcal{A}_{a_w}$, and $\delta$-deviation $\mathcal{A}_{\text{random}}$ in Seaquest, BeamRider and Breakout MDPs with $\delta$-deviation from the optimal trajectory setting with $\delta = 0.003$. The results in Table 3 demonstrate that even a slight change in the

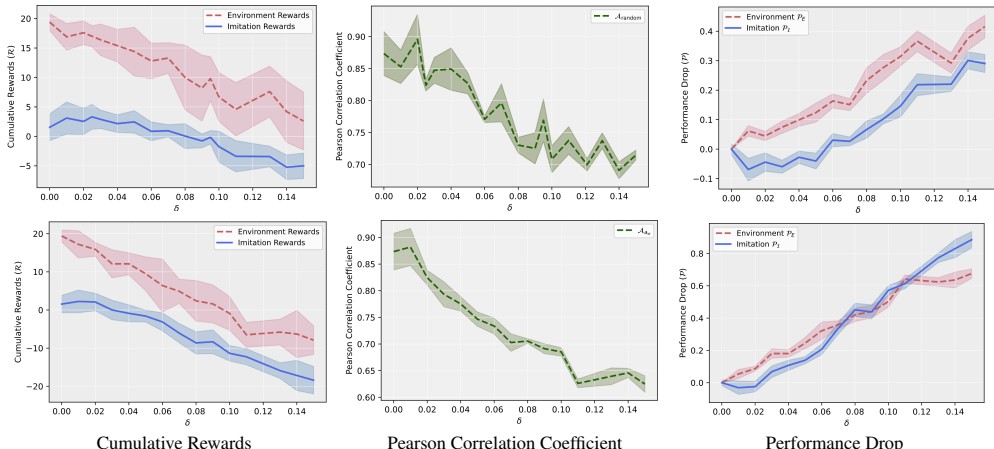

Figure 1: The effects of $\delta$-deviation from the optimal trajectory with $\mathcal{A}_{\text{random}}$ and $\mathcal{A}_{a_w}$. Left: Cumulative rewards obtained from the environment, and inverse $Q$-learning reward predictions. Center: Pearson correlation coefficient. Right: Performance drop based on true environment rewards and inverse $Q$-learning reward predictions. Up: $\delta$-deviation $\mathcal{A}_{\text{random}}$. Down: $\delta$-deviation $\mathcal{A}_{a_w}$.

trajectory (i.e. a change in the trajectory with frequency of 0.003) is more than enough to break the correlation between true rewards obtained from the environment and the reward predictions of the inverse deep neural policy. Hence, without the correlation between true obtained rewards and reward predictions it is evident that the state-of-the-art inverse-$Q$ learning policy cannot be utilized as an inverse reinforcement learning algorithm. These results also correspond well with the theoretical predictions of Corollary 3.4, where the estimated rewards for states that deviate subtly from expert trajectories are uncorrelated with their true rewards. The fact that subtle deviations from the optimal trajectory break the beliefs of the inverse deep neural policy on the MDP rewards, raises significant questions regarding misalignment. The agent's beliefs on the MDP rewards define what the task is and how the task should be solved. The fact that the deep imitation learning policy's beliefs experience an extreme shift under slight deflections from its optimal path is evidence that the policy has a completely different vision on what the objective is and how the task should be solved (i.e. the misalignment problem) (Wiener, 1960; Good, 1965). While these results may raise questions and concerns on the safety and AI-alignment of the deep imitation and deep inverse reinforcement learning policies respectively, one of our main objectives is to layout the exact fundamental trade-off made with learning from expert demonstrations instead of exploring the MDP.

## 6 CONCLUSION

In this paper we study the resilience of policies trained without rewards in high-dimensional state representation MDPs. We essentially seek answers for the following questions: *(i) Does learning without a reward function in complex state representation MDPs cause learning non-robust features independent from the MDP it is trained in and the algorithms it is trained with? (ii) How is the policy robustness affected by the state-of-the-art algorithms that can learn without rewards compared to vanilla trained deep reinforcement learning? (iii) What is the cost of learning from expert demonstrations instead of learning purely from exploration?* To answer these questions we first theoretically motivate that learning from expert demonstrations instead of from pure exploration comes with a cost. Moving along the adversarial directions independent from both the MDP and algorithm in the neural policy manifold we demonstrate that straightforward vanilla trained deep reinforcement learning policies are more robust compared to the state-of-the-art algorithms that can learn without rewards. Following these findings we provide theoretical explanations on the non-robustness of learning without exploration. We further demonstrate that the subtle deviations from the optimal trajectories completely break the correlation between the predicted rewards and the ground truth rewards of the MDP for inverse reinforcement learning policies. Furthermore, we elucidate the relationship between the change in the policy's beliefs on the MDP rewards and the misalignment problem. Most importantly, we highlight that the alignment problem is far beyond being resolved and the algorithms proposed to solve this problem are learning extremely fragile unaligned values.

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
