# OpenReview forum: "Analyzing the Effects of Emulating on the Reinforcement Learning Manifold"
_ICLR.cc/2024/Conference — Submitted to ICLR 2024_

### Official Review · Reviewer_HmFk · 2023-10-31

**Soundness:** 2 fair
**Presentation:** 1 poor
**Contribution:** 2 fair
**Rating:** 3
**Confidence:** 4

**Summary:**

This paper examines the robustness of deep neural policies in adversarial settings. It evaluates the robustness of the deep reinforcement learning algorithm and contrasts it with both imitation learning and inverse reinforcement learning methods. The paper also discusses scenarios in which the optimal trajectory changes due to action perturbations in the policy within the inverse reinforcement learning context.

**Strengths:**

- The domain and motivation are compelling, with a focus on understanding the manifold and assessing minor changes for generalization.
- Attempts have been made to offer both theoretical motivation and empirical evaluation.

**Weaknesses:**

- Throughout the paper, numerous instances of ambiguous language complicate understanding.
- The connections between Lemmas, Propositions, Corollaries, and their proofs to the paper's central message are unclear.
- The experiments are missing crucial details, including specifics on the policy architecture, and the explanations of the results are unclear.

**Questions:**

The paper intends to analyze the reinforcement learning manifold. However, the meaning is not evident from the text. Could you clarify what this entails and how the theoretical and empirical analyses relate to it?

In several sections, the paper mentions "with reward" and "without reward." However, their exact meanings are unclear. Does "without reward" refer to imitation and inverse RL? Furthermore, "without reward" is commonly associated with unsupervised RL, where the agent does not receive a reward signal. Could you please provide clarification? It's crucial to maintain this distinction consistently throughout the paper.

Throughout the paper, several instances of vague language make comprehension challenging. For instance, does “that focus on learning via emulating affect” refer to imitation learning?
And what exactly is meant by "learning from exploration"? Is it referring to regular reinforcement learning?


"Our paper is the first to focus on the adversarial vulnerabilities of deep neural policies that can learn without a reward function." Could you clarify how the proposed method learns without a reward function?


The related work is presented without clearly delineating how the current work differs or aligns with it. Providing these comparisons would enhance the related work section.


While the paper references the general term "deep reinforcement learning," the analysis primarily centers on Q-learning-based approaches. The examination of another form of algorithm, specifically policy gradient, is absent. I suggest the author specify the exact type of algorithm analyzed in the paper for clarity and completeness.

Throughout the paper, I find it challenging to link the various Lemmas, Propositions, Corollaries, and their proofs to the central message of the paper. For instance, the purpose and significance of Lemma 3.1 (and its proof) remain unclear to me. The necessity and meaning behind Propositions 3.3 and 3.5 are also ambiguous. What is the intent of Definition 4.1? How does it relate to the main objective of the paper?

The caption for Table 1 is unclear and requires significant revision. Both Table 1 and Table 2 mention the tasks Pong and Seaquest, yet the results display three different games. How were these results derived?

“The state-of-the-art imitation and inverse reinforcement learning policy is trained via the inverse Q-learning algorithm described in Section 2.” However, I do not see this detailed in the paper. How can both imitation and IRL be trained via Q-learning? The explanation seems to be absent and unclear.

For Figure 1, which environment is being depicted? Is it an average of all three games (Seaquest, BeamRider, and Breakout)? The purpose and message of Figure 1 are unclear.


Overall, the writing needs revision. It predominantly consists of lengthy sentences, making comprehension challenging. Addressing this by crafting shorter, simpler sentences would be beneficial.

Implementation details appear to be missing in the paper. Which architecture is used to represent the policy? Which specific algorithms are utilized for IRL and imitation? Without these specifics, it's challenging to justify the performance presented.

While the paper seems to concentrate on high-dimensional cases, how do the outcomes vary in a low-dimensional representation?

---

> ### Author Response · Authors · 2023-11-14
> **Author Response**
>
> Thank you for the time you have allocated in providing feedback on our paper.
>
> ---
>
> 1. *“In several sections, the paper mentions "with reward" and "without reward." However, their exact meanings are unclear. Does "without reward" refer to imitation and inverse RL?”*
>
> Yes, without reward refers to imitation and inverse RL.
>
> ---
>
> 2. *”Furthermore, "without reward" is commonly associated with unsupervised RL, where the agent does not receive a reward signal. Could you please provide clarification? “*
>
> Yes, as mentioned above without reward refers to imitation and inverse reinforcement learning.
>
> ---
>
> 3. *“For instance, does “that focus on learning via emulating affect” refer to imitation learning?”*
>
> Yes, it refers to imitation learning.
>
> ---
>
> 4. *“What exactly is meant by "learning from exploration"? Is it referring to regular reinforcement learning?”*
>
> Yes, learning from exploration refers to regular reinforcement learning.
>
> ---
>
> 5. *“Could you clarify how the proposed method learns without a reward function?”*
>
> The method mentioned here is the inverse Q-learning algorithm which is also explained in Page 3 between the lines 12 and 19 of the Section titled “Learning without Rewards”.
>
> ---
>
> 6. *“Throughout the paper, I find it challenging to link the various Lemmas, Propositions, Corollaries, and their proofs to the central message of the paper. For instance, the purpose and significance of Lemma 3.1 (and its proof) remain unclear to me. The necessity and meaning behind Propositions 3.3 and 3.5 are also ambiguous.”*
>
> Lemma 3.1, Proposition 3.3, and Proposition 3.5 together theoretically demonstrate  that learning only from expert trajectories results in highly non-robust policies in a natural setting. In particular, Lemma 3.1 proves that states in expert trajectories have large projections onto a subspace implicitly defined by the actions leading to high reward. Proposition 3.3 proves that training on states from the subspace described in Lemma 3.1. results in a policy that is essentially untrained (i.e. takes actions no better than random guessing) outside this subspace.  Proposition 3.5 then uses Proposition 3.3 to analyze a natural setup in which a policy trained in the subspace is highly non-robust. The non-robustness and vulnerabilities of these policies are further justified empirically in Section 4 and Section 5.
>
> ---
>
>  7. *“The paper intends to analyze the reinforcement learning manifold. However, the meaning is not evident from the text. Could you clarify what this entails and how the theoretical and empirical analyses relate to it?”*
>
> The reinforcement learning manifold is represented by the graph of the function $J_{\pi}(s)$, and thus implicitly given by the policy $\pi$.
> Here $J_{\pi}(s)$ is the expected cumulative rewards obtained by policy $\pi$ starting from state $s$.
> The paper both theoretically and empirically analyzes how small perturbations from states $s$ on the learned trajectory yield large changes in $J_{\pi}(s)$ when $\pi$ is trained from expert trajectories, equivalently corresponding to the manifold represented by $J_{\pi}(s)$ having steeper slopes near the learned trajectory.
>
> ---
>
>
> 8. *“What is the intent of Definition 4.1? How does it relate to the main objective of the paper?”*
>
> Definition 4.1 describes the algorithm and MDP independent adversarial direction setting which is a state-of-the-art adversarial framework for producing black-box adversarial perturbations in deep reinforcement learning.
>
> ---
>
> 9. *“Both Table 1 and Table 2 mention the tasks Pong and Seaquest, yet the results display three different games. How were these results derived?”*
>
> The games in the first and fourth row of the Table 1 and 2 represent the MDPs in which the adversarial perturbations are computed. As already described in detail in definition 4.1. the algorithm and MDP independent adversarial direction $\mathcal{A}_{alg+\mathcal{M}}^{random}$ computes an adversarial direction independent from the MDP in which the perturbations are added to the state observations.
>
> ---
>
> 10. *“The state-of-the-art imitation and inverse reinforcement learning policy is trained via the inverse Q-learning algorithm described in Section 2.” However, I do not see this detailed in the paper. How can both imitation and IRL be trained via Q-learning?”*
>
> Please see the **supplementary material** for the training details. Furthermore, see the response to bullet point 5.
>
> ---
>
> 11. *“Implementation details appear to be missing in the paper. Which architecture is used to represent the policy? Which specific algorithms are utilized for IRL and imitation? Without these specifics, it's challenging to justify the performance presented.”*
>
> This is already currently present in the **supplementary material**. Please see supplementary material.

---

> ### Comment · Reviewer_HmFk · 2023-11-21
>
> Thank you, authors, for the response. However, my assessment of the paper, particularly the weaknesses mentioned earlier, remains unchanged.
>
> My concern regarding the implementation details and their impact on the reproducibility of the study persists. In point 11, the author refers to supplementary materials for specifics. While I can find the hyperparameters of the algorithms, there is no mention of the policy architecture. For instance, is a neural network used to represent the Q-network, and if so, how many layers does it have, and what are the dimensions of these layers? Since no code is provided, these implementation details are crucial for replicating the experiments. This is also emphasized by reviewer uuJm, who noted that "...the experimental protocol is not mentioned clearly in the main paper nor the appendix," indicating that the lack of detail presents challenges for reproducibility.
>
> Could you please respond to the following question:
> “For Figure 1, which environment is being depicted? Is it an average of all three games (Seaquest, BeamRider, and Breakout)? The purpose and message of Figure 1 are unclear.”
>
> In point 10, upon reviewing the supplementary materials, I was unable to locate this specific analysis. Could you mention it here or indicate where this detail can be found?

---

> ### Comment · Area_Chair_ujaP · 2023-11-21
>
> Dear Reviewer,
>
> Thank you for your service. The authors have put effort to try to clear all doubts in their response and revision. Can you clarify in details why your assessment remains unchanged?
>
> Best,
>
> AC

---

> > ### Comment · Reviewer_HmFk · 2023-11-22
> >
> > Dear AC (ujaP),
> >
> > I have elaborated on my concerns about the paper in my revised response. I appreciate your attention to the details I initially overlooked. I hope that my expanded feedback will be of use to the author.

---

> > > ### Author Response · Authors · 2023-11-22
> > >
> > > **1.** *”While I can find the hyperparameters of the algorithms, there is no mention of the policy architecture. For instance, is a neural network used to represent the Q-network, and if so, how many layers does it have, and what are the dimensions of these layers?”*
> > >
> > > As also mentioned in the first paragraph of our supplementary material under the Section titled “HYPERPARAMETER DETAILS AND ARCHITECTURES”, the exact same hyperparameters as the original paper [1] are used and reported here. Furthermore, in Line 23 to 24 of our supplementary material it states that vanilla policies are trained with Deep Double Q-Network (DDQN). Hence looking at these lines it is clear that **indeed** neural networks are used to represent the Q-network. It can be seen from both of these citations provided that both of these algorithms use 3 layers of convolutional neural networks of size 32,64,64 respectively.
> > >
> > > [1] IQ-learn: Inverse soft-q learning for imitation. NeurIPS 2021.
> > >
> > > ---
> > >
> > > **2.** *“This is also emphasized by reviewer uuJm, who noted that "...the experimental protocol is not mentioned clearly in the main paper nor the appendix," indicating that the lack of detail presents challenges for reproducibility.”*
> > >
> > > The question Reviewer uuJm asked was also **indeed already in the supplementary material** as well. Please see the response item 1 to the Reviewer uuJm.
> > >
> > > ---
> > >
> > > **3.** *”Could you please respond to the following question: “For Figure 1, which environment is being depicted? Is it an average of all three games (Seaquest, BeamRider, and Breakout)? The purpose and message of Figure 1 are unclear.”*
> > >
> > > It is the average of all three games. The purpose and the message of Figure 1 is **already described** in the paper in Section 5.1.
> > >
> > > ---
> > >
> > > **4.** *“In point 10, upon reviewing the supplementary materials, I was unable to locate this specific analysis. Could you mention it here or indicate where this detail can be found?”*
> > >
> > > This is explained in the supplementary material in Section 1.1 between lines 2 to 4.

---

### Official Review · Reviewer_99sz · 2023-11-01

**Soundness:** 3 good
**Presentation:** 2 fair
**Contribution:** 3 good
**Rating:** 8
**Confidence:** 2

**Summary:**

This work investigates the robustness of vanilla reinforcement learning algorithms compared to methods learning from expert demonstrations, such as inverse reinforcement learning (IRL). The study starts from the formulation of theoretical analysis that shows learning from demonstration can produce lower robustness. In particular, the authors argue that perturbations can cause agents to transition into states where the generated reward from IRL and the states are not correlated, eventually decreasing the agent's robustness by making it not achieve optimal returns in the task of interest. The authors then formulated a way to generate perturbations that cause this highlighted problem and showed vanilla reinforcement learning algorithms yield a lower drop in returns compared to methods learning from expert demonstration.

**Strengths:**

**Originality**

To the best of my minimal knowledge on the topic, the paper appears to be novel. I have not seen other works that establish theoretical analysis on the robustness of inverse reinforcement learning algorithms towards state perturbations.

**Quality - Experiments**

While I did not extensively check the theoretical contributions made in this paper, the experiments that empirically demonstrate the issues with the perturbations proposed in the earlier (theoretical) sections seem to be sound. It believe it empirically demonstrates the claim regarding the robustness issues of IRL methods.

**Clarity**

In general, the paper is well-written. Despite having introduced plenty of theorems throughout the document, the authors did a good job outlining the role of each theorem in highlighting the weaknesses of IRL methods in terms of their robustness. Similarly, I found the experiments (and their analysis) were well written in terms of explaining the overall argument of the paper.

**Significance**

In general, I find that the problem being tackled in this paper could provide highly valuable results for the broader ICLR community. Most reinforcement learning researchers would be highly concerned with the robustness of the policies they trained. While the theoretical analysis seems limited to inverse RL methods, it's still valuable knowledge that perhaps can spur further research in this area. At least, this paper provides the broader RL community with something to consider when choosing between vanilla RL and IRL from expert demonstrations.

**Weaknesses:**

**Clarity - Perturbation used in experiments**

While it may be tricky to produce, it may be useful to show what the perturbations used in the environment really look like. I believe this could help readers further understand the type of "noises" introduced to demonstrate the claims in this paper.

**Questions:**

I do not have further questions.

---

> ### Author Response · Authors · 2023-11-14
> **Author Response**
>
> Thank you for your well-considered and thoughtful review.
>
> ---
>
> *“While it may be tricky to produce, it may be useful to show what the perturbations used in the environment really look like”*
>
> Yes, we will add this information to the supplementary material. Thank you very much for the suggestion.

---

### Official Review · Reviewer_uuJm · 2023-11-09

**Soundness:** 2 fair
**Presentation:** 2 fair
**Contribution:** 2 fair
**Rating:** 5
**Confidence:** 3

**Summary:**

This paper studies the robustness of policies derived from expert demonstrations in the context of high-dimensional Markov Decision Processes (MDPs). The key contribution of this study is the demonstration that deep reinforcement learning policies, trained using the actual reward signal, exhibit much greater robustness against adversarial attacks and perturbations, as compared to policies obtained through inverse reinforcement learning.

The authors further ground their empirical findings with a theoretical framework. They illustrate that in the context of inverse soft-Q-learning with linear policies, the learned rewards are random for states not included within the manifold visited by the expert demonstrations.

**Strengths:**

* The paper studies a relevant problem in light of the current success of RLHF methods and the misalignement problem.
* The findings of the paper clearly point out a substantial lack of robustness of policies learned by IRL to both adversarial attacks and random perturbations.
 * Section 3 offers interesting insights into the shortcomings of linear Q-function approximators in the Inverse Soft Q-Learning setting.

**Weaknesses:**

* The comparisons between the vanilla policies and those derived from IRL are unfair as the vanilla policy is trained on hundreds of thousands of tranisitions whereas its IRL counterpart gets to see only few thousands of transitions.

* The case of studied linear Q-function approximators in the theoretical part is quite restrictive. The assumptions used for Proposition 3.5 are very strong. A potentially more general direction would be to study smooth Q-approximators in the case of Lipschitz MDPs.

* Another weakness is that the authors do not offer no solutions to the problem of lack of robustness. This could be in the form of better training methods for the policy like regularization or better strategies to collect demonstrations.

**Questions:**

I have a few questions for the authors :

1. In the background paragraph, you do not distinguish between Imitation Learning and Inverse Reinforcement Learning. Although IRL covers a broader range of algorithms than can learn even from suboptimal data. Could you please clarify your choice?

2. Could you clarify what you mean by "Manifold setting" in the introduction?

3. Could you clarify what $\phi$ stands for in the inverse Q-learning objective?

4. Relating to the weaknesses, could you provide plots for the evolution of the performance of the IRL policy under adversarial and or perturbation setting in an online setting where it can perform X transitions?

5. I am aware this might not be possible for the current time window. It would've been interesting see a comparison of the robustness of policies learned from expert demonstrations and those learned by ranking of trajectories. As the latter case covers sub-optimal states it can be a potential solution to the robustness problem. Could you add such experiments?

---

> ### Author Response · Authors · 2023-11-14
> **Author Response Part I**
>
> Thank you for dedicating your time to provide kind feedback on our paper.
>
> ---
>
> 1. *”The comparisons between the vanilla policies and those derived from IRL are unfair as the vanilla policy is trained on hundreds of thousands of tranisitions whereas its IRL counterpart gets to see only few thousands of transitions.”*
>
> We would like to kindly highlight that this statement is **false**. The IRL counterpart experiences up to and more than a million transitions.
>
> ---
>
> 2. *“The case of studied linear Q-function approximators in the theoretical part is quite restrictive. The assumptions used for Proposition 3.5 are very strong. A potentially more general direction would be to study smooth Q-approximators in the case of Lipschitz MDPs.”*
>
> An important point here is that the theoretical part is demonstrating a failure mode of training from expert trajectories. Typically this is done by producing a counter-example MDP in which the failure occurs. Thus, a theoretical demonstration of such a counter-example in a simple setting is a stronger result, as it demonstrates that the algorithm fails even in very simple MDPs. MDPs with bounded, linearly parameterized rewards and transitions are immediately Lipschitz, and thus our counter-example setting does indeed correspond to a Lipschitz MDP.
>
> Furthermore, linear function approximation is currently at the frontier of research that develops a rigorous theoretical understanding of reinforcement learning algorithms [1,2,3,4,5,6,7,8]. Thus, our theoretical results are demonstrated in the most general setting currently amenable to provable mathematical analysis.
>
> [1] Bilinear classes: A structural framework for provable generalization in RL. ICML, 2021.
>
> [2] Reinforcement learning from partial observation: Linear function approximation with provable sample efficiency. ICML 2022.
>
> [3] Provably efficient reinforcement learning with linear function approximation. Conference on Learning Theory. PMLR, 2020.
>
> [4] Nearly minimax optimal reinforcement learning for linear mixture markov decision processes. Conference on Learning Theory. PMLR, 2021.
>
> [5] Learning near optimal policies with low inherent bellman error. ICML, 2020.
>
> [6] Reinforcement learning in linear MDPs: Constant regret and representation selection. NeurIPS 2021.
>
> [7] Nearly minimax optimal reinforcement learning with linear function approximation. ICML, 2022.
>
> [8] First-order regret in reinforcement learning with linear function approximation: A robust estimation approach. ICML 2022.

---

> ### Author Response · Authors · 2023-11-14
> **Author Response Part II**
>
> ---
>
> 3. *“Another weakness is that the authors do not offer no solutions to the problem of lack of robustness. This could be in the form of better training methods for the policy like regularization or better strategies to collect demonstrations.”*
>
> Please note that in the adversarial machine learning literature there is a line of work dedicated solely to discussing the explicit and concrete vulnerabilities of the models [1,2,3,4,5,6,7], and there is a line of work dedicated to addressing these problems described in these studies [8,9]. We believe these two lines of research progress side by side, without the expectation on one single paper carrying both of these perspectives at the same time.
>
>
> [1] On Adaptive Attacks to Adversarial Example Defenses, NeurIPS 2020.
>
> [2] Label-Only Membership Inference Attacks, ICML 2021.
>
> [3] Poisoning and Backdooring Contrastive Learning, ICLR 2022.
>
> [4] Adversarial Robust Deep Reinforcement Learning Requires Redefining Robustness, AAAI 2023.
>
> [5] EAD: Elastic-Net Attacks to Deep Neural Networks via Adversarial Examples, AAAI 2018.
>
> [6] Deep Reinforcement Learning Policies Learn Shared Adversarial Features Across MDPs, AAAI 2022.
>
> [7] Stealthy and Efficient Adversarial Attacks against Deep Reinforcement Learning, AAAI 2020.
>
> [8] Robust Deep Reinforcement Learning against Adversarial Perturbations on State Observations, NeurIPS 2020.
>
> [9] Robust Deep Reinforcement Learning through Adversarial Loss, NeurIPS 2021.
>
> ---
>
> 4. *“Could you clarify what you mean by "Manifold setting" in the introduction?”*
>
> Manifold settings here refers to the first paragraph of the introduction describing the current fields that reinforcement learning is being utilized in, such as autonomous vehicles, finance and large language models.
>
>
> ---
> 5. *“Could you clarify what $\phi$ stands for in the inverse Q-learning objective?”*
>
> $\phi:\mathbb{R}\to\mathbb{R}$ is a concave function corresponding to a choice of distance metric for regularization in inverse Q-learning. For example, setting $\phi(x) = x$ corresponds to total variation distance, and setting $\phi(x) = x - x^2/4$ corresponds to $\chi^2$ distance. Please see [1] for more details.
>
> [1] IQ-Learn: Inverse soft-Q Learning for Imitation, NeurIPS 2021.
>
> ---
>
> 6. *“Relating to the weaknesses, could you provide plots for the evolution of the performance of the IRL policy under adversarial and or perturbation setting in an online setting where it can perform X transitions?”*
>
> Please see the response to bullet point 1.
>
> ---
>
> 7. *”I am aware this might not be possible for the current time window. It would've been interesting see a comparison of the robustness of policies learned from expert demonstrations and those learned by ranking of trajectories. As the latter case covers sub-optimal states it can be a potential solution to the robustness problem. Could you add such experiments?”*
>
> In the RLHF setting, the trajectories selected for ranking are the outputs of an LLM (e.g. GPT-4), and hence these trajectories analogously are the “expert trajectories”. Thus, the analysis of our paper extends to the setting of ranking trajectories. Furthermore, note that LLMs have had hard limits applied to the number of rounds of dialogue with users, due to the observations that unlimited conversations steer towards harmful behavior by the LLM (i.e. insulting users, manipulating, or lying) [1].
>
> [1] New York Times. Microsoft to Limit Length of Bing Chatbot Conversations, 2023.

---

> > ### Comment · Reviewer_uuJm · 2023-11-21
> >
> > We thank the authors for their response. However i have some doubts regarding some claims.
> >
> > > ”The comparisons between the vanilla policies and those derived from IRL are unfair as the vanilla policy is trained on hundreds of thousands of tranisitions whereas its IRL counterpart gets to see only few thousands of transitions.”
> > We would like to kindly highlight that this statement is false. The IRL counterpart experiences up to and more than a million transitions.
> >
> > This point deserves further clarification. Indeed, the experimental protocol is not mentioned clearly in the main paper nor the appendix.  Indeed, it is not stated clearly if the experiments follow an offline or online setting for IQ-L. How many expert trajectories and online samples are used.
> >
> > According to the IQ-learn paper, for the Atari games, 20 expert trajectories are used supplemented by up-to 1 Million online interactions with the environment. This is still a magnitude lower than the tens of millions of trajectories used in the Atari games [1].Hence, it would be worth mentioning the number of samples used for the vannila RL method and that used by IQ-Learn. Or even better to perform an ablation on the robustness of IQ-L as a function of the number of online samples.
> >
> > The experimental section should be rewritten to better clarify the protocol used for the DQN and IQ-Learn baselines.
> >
> > > ”I am aware this might not be possible for the current time window. It would've been interesting see a comparison of the robustness of policies learned from expert demonstrations and those learned by ranking of trajectories. As the latter case covers sub-optimal states it can be a potential solution to the robustness problem. Could you add such experiments?”
> > In the RLHF setting, the trajectories selected for ranking are the outputs of an LLM (e.g. GPT-4), and hence these trajectories analogously are the “expert trajectories”. Thus, the analysis of our paper extends to the setting of ranking trajectories. Furthermore, note that LLMs have had hard limits applied to the number of rounds of dialogue with users, due to the observations that unlimited conversations steer towards harmful behavior by the LLM (i.e. insulting users, manipulating, or lying) [1].
> >
> > It is not clear how the analysis of the paper extends to rankings of trajectories. Also in a general RL setting like a locomotion task or Atari games, one can imagine having access trajectories of varying performance and their respective rankings. Such trajectories could potentially cover more effectively the state space of MDP and exhibit more robustness. Indeed, in the experimental section, it is only normal for the agent to fail once it visits a single sub-optimal state as it has never encountered such states during training.

---

> ### Author Response · Authors · 2023-11-22
>
> **1.** *”Indeed, the experimental protocol is not mentioned clearly in the main paper nor the appendix…How many expert trajectories and online samples are used?...The experimental section should be rewritten to better clarify the protocol used for the DQN and IQ-Learn baselines.”*
>
> This is indeed mentioned in our supplementary material. Please see Line 14 (i.e. item 9), and Line 18 respectively of the Section Titled “HYPERPARAMETER DETAILS AND ARCHITECTURES” of the supplementary material.
>
> ---
>
> **2.** *”It is not clear how the analysis of the paper extends to rankings of trajectories. Also in a general RL setting like a locomotion task or Atari games, one can imagine having access trajectories of varying performance and their respective rankings. Such trajectories could potentially cover more effectively the state space of MDP and exhibit more robustness. Indeed, in the experimental section, it is only normal for the agent to fail once it visits a single sub-optimal state as it has never encountered such states during training.”*
>
> Ranking trajectories requires human feedback (as in RLHF training for LLMs). Thus, increased coverage of the state-space via ranked trajectories comes at a very significant human labor cost, this is a substantially different level of cost than only having a few expert demonstrations, as used in the IQ learn algorithm we study.

---

### Meta-Review · Area_Chair_ujaP · 2023-12-05

**Metareview:**

This paper focuses on the non-robustness of the state-of-the-art algorithms that accomplish learning without rewards in high dimensional state representation MDPs.

**Reviewers have reported the following strengths:**

- The studied problem is relevant;
- Interesting findings on the lack of robustness of established methods;

**Reviewers have reported the following weaknesses:**

- Quality of writing;
- Empirical evaluation.

**Decision**

This paper received a positive evaluation with low confidence from one Reviewer, and two negative assessments from the others. Despite the authors put good effort into the rebuttal, the Reviewers have not changed their assessment. In particular, the outcome of the proposed analysis was not clear to most Reviewers, due to low quality of writing and presentation. Moreover, some Reviewers pointed out the absence of several hyperparameters from the main paper and the appendix; the authors clarified that all this information is in the appendix. However, the Reviewers did not reply to the authors nor acknowledged the presence of this information. It seems to me that most of the requested hyperparameters are actually in the appendix; thus, I decided to downweigh this issue for the final decision. Still, I consider the quality of the writing and presentation insufficient, and I strongly encourage the authors improving their work in a future submission.

**Justification For Why Not Higher Score:**

N/A

**Justification For Why Not Lower Score:**

N/A

---

### Decision · Program_Chairs · 2024-01-16

Reject